# Management of Locally Advanced Laryngeal Cancer—From Risk Factors to Treatment, the Experience of a Tertiary Hospital from Eastern Europe

**DOI:** 10.3390/ijerph20064737

**Published:** 2023-03-08

**Authors:** Anca-Ionela Cîrstea, Șerban Vifor Gabriel Berteșteanu, Răzvan-Valentin Scăunașu, Bogdan Popescu, Paula Luiza Bejenaru, Catrinel Beatrice Simion-Antonie, Gloria Simona Berteșteanu, Teodora Elena Diaconu, Petra Bianca Taher, Simona-Andreea Rujan, Irina-Doinița Oașă, Raluca Grigore

**Affiliations:** 1Department 12-Otorhynolaryngology, Ophtalmology, Faculty of Medicine, “Carol Davila” University of Medicine and Pharmacy, 050474 Bucharest, Romania; anca-ionela.cirstea@drd.umfcd.ro (A.-I.C.);; 2Department of ENT, Head and Neck Surgery, Colţea Clinical Hospital, 030167 Bucharest, Romania; 3Department of General Surgery, Colţea Clinical Hospital, 030167 Bucharest, Romania; 4Department of ENT, Head and Neck Surgery, “Carol Davila” Emergency Central Military Hospital, 010825 Bucharest, Romania

**Keywords:** laryngeal cancer, risk factors, advanced cancer

## Abstract

Laryngeal cancer is an important oncological entity in which prognosis depends on the establishment of appropriate preventive and diagnostic measures, especially in high-risk populations. We present a retrospective two-year study (January 2021 to December 2022) with 152 patients diagnosed with laryngeal cancer from a tertiary hospital in Romania. The average age of the patients was 62 years old for both sexes, with a range from 44 to 83 years. The most frequent symptom was dysphonia with or without dyspnea in 142 cases (93.42%), followed by dyspnea alone in nine patients (5.92%) and dysphagia in one case (0.66%). Surgical treatment in this study consisted of partial laryngectomy (CO_2_ laser transoral tumor ablation, supraglottic horizontal laryngectomy or hemilaryngectomy), or total laryngectomy. The main treatment was total laryngectomy (63%). For the eight patients with initial organ preservation treatment, the average time of recurrence was about two-and-a-half years. For the four patients who underwent a total circular pharyngo-laryngectomy, the upper digestive tract needed to be rebuilt with a salivary bypass tube or with a tubed myocutaneous flap from the major pectoralis muscle. One strong point is characteristic of the study group in gathering patients with advanced stages of laryngeal carcinoma candidates for salvage surgery and extended reconstruction methods. The development of new prevention protocols is mandatory in Eastern European countries.

## 1. Introduction

Laryngeal cancer is an important oncological entity in which prognosis depends on the establishment of appropriate preventive and diagnostic measures, especially in high-risk populations. There has been an upward trend in the incidence of larynx cancer over the last three decades, with Central and Eastern Europe being the top geographies, according to Globocan [1].

Smoking and heavy alcohol consumption are the main risk factors associated with larynx cancer. In addition, aging, HPV infection, exposure to paint, asbestos, gasoline fumes, and radiation play an important role [2].

Nowadays, although diagnostic methods have evolved, many patients still present advanced disease stages during diagnosis and require total laryngectomy. This surgical treatment has an important psychophysical and social impact on the quality of life of patients. This is caused by functional changes that result from laryngeal removal, with an immediate loss of speech function and altered respiratory physiology. In addition, there are different approaches for advanced laryngeal cancer and recurrent disease with locoregional metastasis. The main goals of locally advanced laryngeal cancer treatment are the local control of the disease, survival, and, if possible, the preservation of laryngeal anatomy and functions (speech, swallowing, and airway patency). The debate about the ideal treatment for laryngeal cancer is old and still ongoing. The historical gold standard for advanced laryngeal tumors is total laryngectomy. Although this procedure is still the treatment of choice for advanced laryngeal cancer, more conservative treatments such as radiotherapy, chemotherapy, targeted molecular therapy, and organ-preserving surgery are becoming more and more frequently used to achieve organ preservation [3]. There are many eligibility criteria for larynx preservation strategies that are aimed at obtaining a positive treatment outcome. A low irradiation tolerance can cause discontinuation or even the stopping of radiotherapy altogether, and low chemotherapy tolerance requires dose reduction. In both cases, the patient receives a suboptimal treatment that is responsible for a lower tumor response: a situation frequently associated with salvage laryngectomy [4].

The treatment options and the type of surgery in the advanced stages are guided by several factors. If the prevertebral fascia or carotid artery is invaded, the case is declared unsuited for surgical removal. In situations such as these, systemic therapy will be chosen [5].

Tumor extension to the pharynx or esophagus may require a total circular pharyngo-laryngectomy and the reconstruction of the upper digestive tract continuity with the myocutaneous flap from the pectoralis major muscle or another reconstructive method.

The aim of the study was to describe the main risk factors found in laryngeal cancer patients and the treatment methods and outcomes for advanced stages, including reconstruction techniques from a referral oncological center in Eastern Europe. The incidence rates of laryngeal cancer in this region are among the highest recorded in Europe. As early detection strategies and the implementation of effective laryngeal cancer prevention must be based on accurate and reliable information, this article provides a broad experience with advanced laryngeal cancer treatment and its outcomes from a tertiary hospital in Romania, Eastern Europe.

## 2. Materials and Methods

This retrospective study analyzed the patients diagnosed with larynx cancer between January 2021 and December 2022 from Colțea Clinical Hospital, Bucharest. Only patients with stage III or IV laryngeal cancer who came to our hospital during the above-specified time span were included in this study. The workup included a complete head and neck examination, fiberoptic examination, a neck CT scan with contrast, an MRI with contrast in nine selected cases due to the suspicion of invasion into the prevertebral fascia, and a chest CT scan. The biopsy was performed through direct laryngoscopy under general anesthesia. All the patients’ cases were evaluated by the hospital’s multidisciplinary tumor board, and the cases with advanced laryngeal cancer were included in this study. The patients who had distance metastases at the time of diagnosis were excluded from the study. The patients who qualified for surgery had preanesthesia evaluation performed.

Both primary tumors and recurrences were enrolled in this study. The patient’s history included the documentation and quantification of tobacco and alcohol use.

The patients were classified as having stage III and stage IV laryngeal cancer, according to the Union for International Cancer Control (UICC)/American Joint Committee on Cancer (AJCC) staging [6]. The advanced stages groups also included patients with an early T classification (T1/2). However, the criteria for the advanced stage were the presence of nodal disease (N2-3).

Definitive treatment options (surgery, radiotherapy, chemoradiotherapy, or a combination of these) for every case were discussed at the tumor board meeting. The final treatment choice was based on each patient’s preference.

A database with all the information regarding demographic and medical data was created. We used Microsoft Excel 16.60 and Numbers (Mac IOS) to process the data.

## 3. Results

### 3.1. Epidemiological Data

Within two years (January 2021–December 2022), 152 patients were diagnosed with advanced laryngeal neoplasms in the Otolaryngology Clinic of Colțea Clinical Hospital Bucharest. Most of the patients were males (87%), with slight domination for the non-urban area (82 rural vs. 50 urban). The average age was 62 years old for both sexes, with a range from 44 to 83 years and the highest percentage in the 60+ age group (61%) (Figure 1).

The average female age was 52.8, with a range from 44 to 63. The youngest participant in the study was 44. (Figure 2). All female patients were diagnosed with stage IV.

### 3.2. Risk Factors

Tobacco smoking is the most significant risk factor associated with laryngeal cancer. According to the definitions from the Centers for Disease Control and Prevention, the current smoker is defined as an adult who has smoked at least 100 cigarettes in his or her lifetime and if he/she is still smoking at the time of the interview. The former smoker is an adult who has smoked at least 100 cigarettes in his or her lifetime but who has quit smoking at the time of the interview. The never smoker is an adult who has never smoked or who has smoked less than 100 cigarettes in his or her lifetime. Regarding the number of cigarettes smoked per day, a heavy smoker is, according to the recommendations of the World Health Organization, a smoker with a daily cigarette consumption of over 20 pieces. Analyzing the data, our patients were 83% current smokers, 15% former smokers, and only 2% never smokers. Figure 3 shows that from the total of 126 current smokers, 106 were severe smokers, and 20 were mild smokers. Of the total 23 former smokers, 17 were severe smokers, and six were mild smokers [7].

In this study, we considered only classical nicotine cigarette smoking. The use of electronic cigarettes is not very common in Romania. It is also much more expensive, so none of our patients smoked electronic cigarettes.

Alcohol consumption was reported as positive by 78.2% of the patients, 11.8% denied it, and we found a lack of consignment in 10% of the patients.

Regarding the association between alcohol and smoking, 81.1% of the patients participating in our study were in this high-risk group. All female patients from this study were former smokers (25%) or nonsmokers (75%).

All the patients were instructed to quit tobacco and alcohol after initiating any type of treatment. Since the study patients were from groups of social risk and lived in country-side areas, they did not regularly visit general practitioners, so we suspect that some of them continued to use tobacco and alcohol after radiation therapy. If we consider their answers at regular visits, they said that they quit smoking and drinking alcohol after starting radiation therapy. Based on the answers from their families, 9% of the patients continued to smoke, 6% of the patients continued to drink, and 4% continued to use both tobacco and alcohol.

Eight patients were previously exposed to radiotherapy for laryngeal cancer in order to preserve the organ. Two of these patients had a previous partial laryngectomy and then underwent radiotherapy and systemic chemotherapy. Six patients only had concomitant radiotherapy and chemotherapy. We noted the differences in radiotherapy technique outcomes while neglecting patients’ general status, and we think this is an explanation for why most surgeons prefer total laryngectomies as the treatment of choice.

Exposure to other chemicals (asbestos, chromium, nickel) incriminated in laryngeal oncogenesis was not specified in our patients’ history. A total of 89% of the patients had poor oral health with numerous untreated dental caries. Gastroesophageal reflux disease symptoms were also present in 21% of the patients.

Only one patient had a family history of laryngeal cancer.

### 3.3. Symptoms and Tumor Description

As we can see in Figure 4, the most frequent symptom was dysphonia with or without dyspnea in 142 cases (93.42%), followed by dyspnea alone in nine patients (5.92%) and dysphagia in one case (0.66%). The mean lag time between the onset of symptoms and the first consultation was 12 months (three months–two years). 

Regarding tumor localization [International Classification of Diseases 11th Revision (ICD-11)], about 63% were malignant neoplasm of the glottis (C32.0), followed by supraglottis 30% (C32.1), and subglottis 7% (C32.2) (Figure 5). According to AJCC staging, most of the tumors were stage IV, respectively 92 cases (Figure 6). All patients were M0. From the histopathological perspective, almost all the tumors were non-keratinized and keratinized squamous cell carcinomas (about 96.2%). The other types of tumors identified were verrucous carcinoma and acantholytic carcinoma.

Based on TNM classification, in the cases of partial laryngectomy, four patients were T3, four patients were T4, and 12 patients were T1-T2 with N2-N3 nodal disease. Of the patients that underwent total laryngectomy, 24 patients were T3, 44 patients were T4, and 28 patients were T1-T2 with N2-N3 nodal metastases. The four cases with total circular pharyngo-laryngectomy were T4. The pathological TNM classification consisted of 38 pT4 patients, 42 pT3 patients, 34 pT2 patients, and six pT1 patients. Regarding the nodal metastases, eight patients were classified as pN1, 43 as pN2, and 69 as pN3. The four patients referred to radiation therapy and chemotherapy were T3, 36 were T4, and eight patients were T1 or T2 but with nodal metastases N2-N3.

### 3.4. Treatment Options

Surgical treatment in this study consisted of partial laryngectomy (CO2 laser transoral tumor ablation, supraglottic horizontal laryngectomy, or hemilaryngectomy) or total laryngectomy. In two of the cases, total laryngectomy with a total thyroidectomy was performed, and in four cases, a total circular pharyngo-laryngectomy was performed.

The main treatment in our study was total laryngectomy (63%, *n* = 96). For all these patients, bilateral neck dissection was performed as well. The criteria for total laryngectomy were vocal fold fixation, the invasion of the postcricoid area, invasion through the thyroid cartilage, or the invasion of the tissue beyond the larynx.

A phonatory fistula between the trachea and esophagus was carried out in 63 cases with prosthesis positioning through a primary puncture. A secondary puncture was performed in 12 cases. Elective neck dissection was performed for patients with a clinically N0 neck. Patients who underwent primary non-surgical treatment did not undertake neck dissection. A nasogastric feeding tube was used during the first two weeks following the surgery. Subsequently, a blue dye test was performed on the patients in order to determine if there was a pharyngocutaneous fistula.

Despite the fact that 75 of the patients with total laryngectomy had vocal prosthesis mounted, 34% of them did not use it at all.

The maximum time for voice and deglutition rehabilitation was six weeks for the case with a total laryngectomy and 11 weeks for the case with a partial laryngectomy.

In Figure 7, we can see that in the group with RT and system therapy, the majority of the patients were in stage IV. A total of 12 patients who underwent a partial laryngectomy had a small laryngeal tumor (T1-T2) but N2-3 neck metastases. In 16 patients, the tumor had an extension to the posterior pharyngal wall, with the invasion of the prevertebral fascia. As a result, the patients were referred to radiotherapy and chemotherapy.

Four patients refused total laryngectomy and underwent concurrent systemic therapy and radiotherapy. Two patients died before starting any type of treatment. Another two patients had high-risk factors for anesthesia, so they underwent radiotherapy.

In our study, total laryngectomy has a constant rate between the ages of 40 and 60, growing constantly towards 80 years old, while RT+CHT remains linear in all age groups (Figure 8).

All patients undergoing surgery received radiation therapy at six–eight weeks postoperatively. Concurrent radiotherapy and systemic therapy were administered to patients with adverse pathologic features represented by extranodal extension, positive margins, close margins, pT4 primary, pN2 or pN3 nodal disease, perineural invasion, vascular invasion, and lymphatic invasion. The decision to enhance radiotherapy with chemotherapy depended on the patient’s general condition, medical co-morbidity, and ability to tolerate chemotherapy. The postoperative radiotherapy protocol consisted of 60–66 Gy (2.0 Gy/fraction) received daily from Monday to Friday for six-six-and-a-half weeks.

Patient cases that were unsuited for surgical tumor removal received two cycles of Cisplatin and 5-Flurouracil. Patients who did not tolerate this regimen due to medical reasons received alternative agents, often with carboplatin or cetuximab. All patients completed a full course of standard fractionated radiotherapy, which was administered five days a week for seven weeks. The target dosage for the primary tumor was 66 Gy (2.2 Gy/fraction)–70 Gy (2.0 Gy/fraction). Bilateral lymph nodes (levels 2–4) were also included in the radiation field.

Both primary tumors and recurrences were enrolled in the study. We enrolled eight patients with recurrence after previous treatment. Two of these patients had a previous partial laryngectomy and then underwent radiotherapy and systemic chemotherapy. Six patients only had concomitant radiotherapy and chemotherapy.

For the eight patients with initial organ preservation treatment who were enrolled in our study with recurrence, the average time that recurrence arose was about two and a half years. All recurrence cases were confirmed with direct laryngoscopy and biopsy. They were restaged with a CT scan or positron emission tomography CT for more accuracy. All these patients underwent salvage laryngectomy and bilateral neck dissection.

For the four patients who underwent a total circular pharyngo-laryngectomy, the upper digestive tract needed to be rebuilt. In two cases the reconstruction was performed with a Montgomery salivary bypass tube, and in another two cases a tubed myocutaneous flap from the major pectoralis muscle was used (Figure 9a,b).

## 4. Discussion

In our series, we found an average age of 62 years (from a range of 44 to 83); this is slightly younger than the average age found in another recent study (Koroulakis et al.—65 years) [2]. Taking into account the sex of patient cases, we presented a 6.6:1 ratio, confirming the association of risk with the male sex, similar to the findings reported by Nocini et al., with 5:1 male/woman [8]. This is likely attributed to the frequent addiction habits among males.

The vast majority of the participants were males and heavy smokers. In Romania, there has been a descending trend line over the last 20 years (Table 1); however, the incidence of laryngeal cancer is still high. According to Eurostat, the smoking rate in Europe was 18.4% in 2019 compared to 28.4% in Romania [9].

In our study group, the most frequent symptom was dysphonia with or without dyspnea in 142 cases (93.55%), followed by dyspnea in nine patients (5.92%) and dysphagia in one patient (0.66%). Our findings are similar to the data reported by Herrera-Goméz et al., where the symptoms were dysphonia in 458 patients (91.6%), dysphagia in 110 patients (22%), dyspnea in 103 patients (20.6%), odynophagia in 37 patients (7.4%), and bleeding in 11 patients (2.2%) [10]. Dechaphunkul et al. referred to dysphonia in 97.2% of the patients included in their study, accompanied by dyspnea in 24.4% of cases and odynophagia in 13.3% [11]. The first symptoms are well-known by otorhinolaryngologists but ignored both by patients and primary care. If all specialties recognized the first symptoms and knew the impact of surgical and radiotherapy treatment, the disease could be better managed. There is a gap between internal medicine, other surgical specialties, and general practitioners, who do not know the dimensions of oncological treatment for head and neck cancer as well as the impact on QOL. Utilizing accurate information from specialized clinics could have a major impact on diagnosing the disease in its early stages, as well as developing screening programs, as there would be an increased awareness of head and neck cancer. Taking into account the direct relationship between a risk factor and the impact of the disease, together with the existent information in the field from other European countries and globally, is key in developing good healthcare programs and prevention. This will decrease the costs of disease management as well.

The histopathological result was squamous cell carcinoma in 96.2% of the cases, similar to those reported in the literature [12,13].

A discordance between cTNM and pTNM was observed in the cases with prior radiotherapy, with the clinical T being overestimated. This could be the result of fibrosis caused by radiation therapy and the misinterpretation of computer tomography images.

Locally advanced cancers, including T3-4 and N1-3 disease, are more difficult to treat and usually require combination therapies. Although these cancers are not candidates for laryngeal preservation surgery if surgically resectable, definitive radiation therapy in association with cisplatin chemotherapy remains an option to organ preservation surgery [14]. According to Dyckhoff et al. [15], in advanced laryngeal cancer, organ preservation with primary conservative treatment is likely to result in a significantly worse outcome in terms of overall survival, and the patients should be informed about this.

Despite the fact that The American Society of Clinical Oncology gives us practical clinical instructions on the application of Clinical Guidelines for the Use of Larynx-Preservation Strategies in the Treatment of Laryngeal Cancer, in our study, the main treatment was total laryngectomy (*n* = 96, 63.15%) [16]. The reality of the Eastern European countries in oncological treatments is heavily influenced by:–Late diagnosis;–High costs for oncological treatments, chemotherapy, and radiotherapy;–A small number of new techniques for radiotherapy;–A small number of centers/hospitals and specialists;–No protocols according to international standards approved by authorities.

Four patients who were candidates for total laryngectomy refused the intervention and underwent radio chemotherapy.

Bilateral neck dissection was also performed in all the patients with salvage laryngectomy from our study. According to Basheeth et al. [17], bilateral neck dissection during salvage laryngectomy has been reported to have a high incidence of major complications, including pharyngocutaneous fistula.

The reconstructive methods used in the cases of extended tumor resection were represented by the Montgomery bypass salivary tube and major pectoralis muscle pedicled flap for the upper digestive tube reconstruction. Other methods that have been described are represented by gastropharyngeal anastomosis, radial forearm free flap, and anterolateral thigh free flap. The bypass salivary tube reconstruction procedure was preferred due to the short surgical time, the shorter hospitalization stays, and lower associated costs. It also allowed for the success of radiotherapy.

Salvage surgery after non-surgical treatment is known to have up to 50% higher complication rates than primary surgery with an incidence of pharyngocutaneous fistula. In the literature, the major pectoralis muscle pedicled flap is often used after the salvage of the total laryngectomy in order to prevent the development of a pharyngocutaneous fistula [18]. No pectoralis major flap was performed on any of our salvage total laryngectomy cases, and this might have been the surgeons’ decision.

Most of the patients from our study were current smokers (83%), a significant proportion (15%) were former smokers, and only 2% never smoked. In the study conducted by Menach et al. [19], the results suggest that being a former smoker confers a positive risk for laryngeal cancer across all laryngeal subsites, the highest being for supraglottic cancer (OR = 6.7), which is comparable to glottic carcinoma (OR = 6.1) when compared to the controls. Lewis, his team, and other authors demonstrated that the glottis is anatomically the narrowest part of the upper airway, and for that reason, it is more susceptible to the deposition of inhaled carcinogens found in cigarette smoke [20,21,22,23,24]. The delays in the implementation of tobacco preventive measures in countries from Eastern Europe, compared to Western Europe, are mainly responsible for the differences in mortality trends between these areas [25].

All female patients from this study were former smokers (25%) or nonsmokers (75%), with an age range from 44 to 63, which is younger than the age described in other studies. In our study, predominant stage for the female patients was T4, unlike the literature, where they were usually present in earlier stages [26]. We partially explained this fact by the lack of education and screening measures. Other factors described in the literature that are associated with laryngeal cancer are represented by alcohol consumption and particular dietary aspects [27]. The significant difference between the male-to-female ratio and the low incidence of smoking among female patients suggests that other factors in laryngeal cancer development might play a role, such as hormonal influences. Other study found that there is compelling evidence that laryngeal cancer is hormone-responsive, specifically to the effects of E2 via Erα36 [28]. Further studies should seek a clearer understanding of the factors involved in female laryngeal cancer.

Abnormally expressed genes in laryngeal squamous cell carcinoma have been studied in recent years. Such genes are represented by *LC35C1*, *HOXB7*, and *TEDC2,* and they have the potential to become new therapeutic targets and prognostic biomarkers for laryngeal squamous cell carcinoma [29].

The majority of the patients from our study (89%) had poor oral hygiene. In the literature, it is described that oral health has a direct effect on tumor biology due to the associated immune or inflammatory response. In the study of Hashim et al., laryngeal cancer was inversely associated with <5 missing teeth, brushing ≥once/day, regular dentist visits, and wearing dentures [30,31].

A study conducted by Hemilton et al. [32], which included 114 participants, found that chemoradiation was preferred by 62% of the surveyed participants, over laryngectomy, in contrast to our study, where total laryngectomy was performed in 63% of the cases.

The disease-free survival rate for eight patients in our study who underwent organ preservation treatment was two and a half years. In the European study of Lefebre et al. [33], the disease-free survival at five years was 25% for chemotherapy and 27% for immediate surgery. In another study, the global five-year disease-free survival was 47.5% (38% in patients undergoing radiotherapy only, 49% when chemotherapy was added, and 45% in cases with induction chemotherapy for chemoradiotherapy) [34].

We found that although speech prostheses were inserted in almost all patients with total laryngectomy, 34% of them did not use them. The insurance system pays only for two prostheses per year, and sometimes a dysfunctional valve could lead to therapy failure. Social reintegration was much faster for patients who learned to use the phonatory prosthesis. The resumption of swallowing and phonation in partial laryngectomy cases was successfully completed in a maximum timeframe of six months: significantly longer compared to those who had total laryngectomy (11 weeks).

The reconstruction procedure with a bypass salivary tube was applied to all patients due to low costs and the lower rate of post-radiotherapy complications relating to nutrition. The main disadvantage is the impossibility of using the phonatory prosthesis for voice rehabilitation.

The impact of head and neck cancer is one of the largest in all malignancies. Early-stage diagnosis is the only method for organ preservation and QOL. Continuous epidemiological surveillance based on data from accurate and reliable sources should be considered the cornerstone for increasing preventive and early diagnostic interventions in categories of patients with a higher risk for this type of malignancy. It is imperative to start awareness programs and develop prevention protocols in Eastern European countries. Preventive protocols should include patient education regarding dental caries prevention, diet counseling, and meticulous oral hygiene instructions.

## 5. Conclusions

Our study confirmed that the main risk factors associated with laryngeal cancer are tobacco smoking and alcohol consumption. The main treatment for advanced stages in European developing countries is still total laryngectomy (63% of patients). Women are affected at a younger age, and the association with smoking is less frequent than in the case of men.

One strong point is characteristic of the study group in gathering patients with advanced stages of laryngeal carcinoma candidates for salvage surgery and extended reconstruction methods.

Therefore, knowing the peculiarities of different regions of the world, we need to develop better prevention methods, improved screening programs, and increase knowledge among healthcare professionals.

## Figures and Tables

**Figure 1 ijerph-20-04737-f001:**
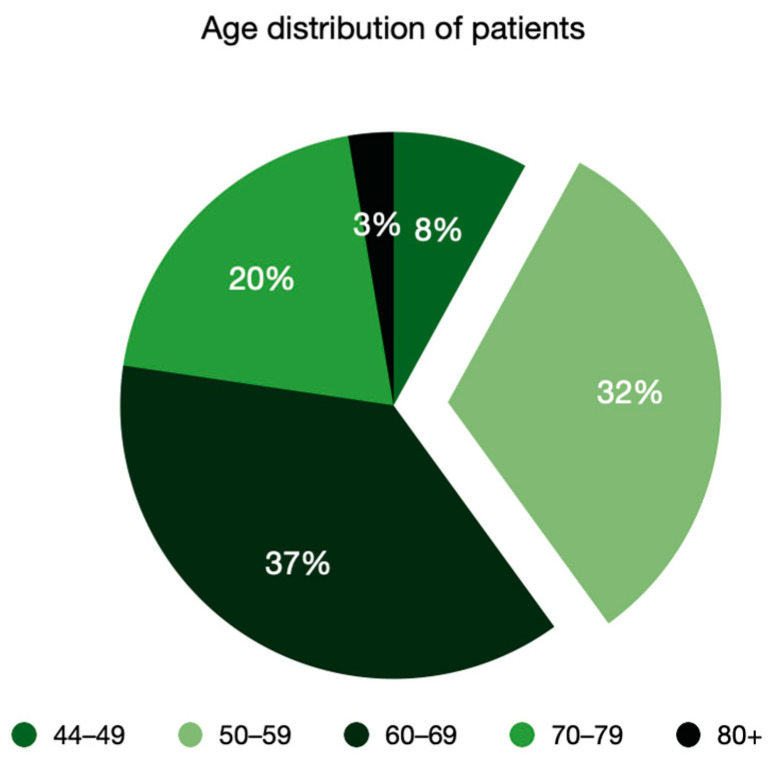
Age distribution.

**Figure 2 ijerph-20-04737-f002:**
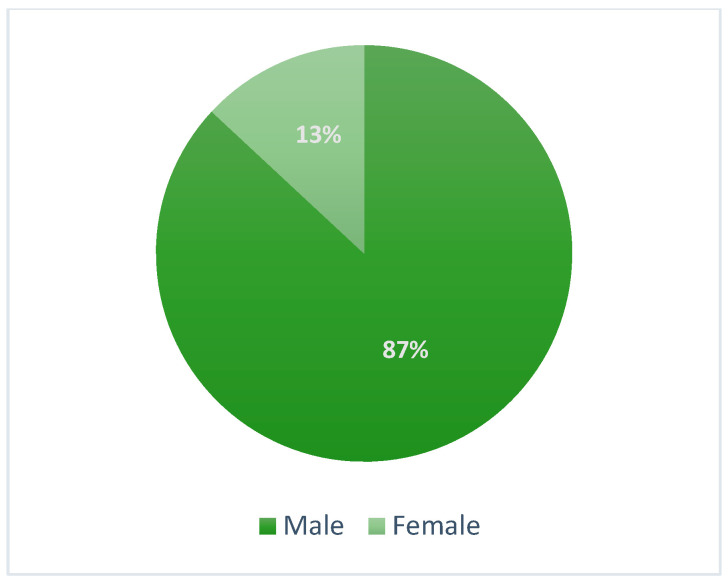
Sex distribution.

**Figure 3 ijerph-20-04737-f003:**
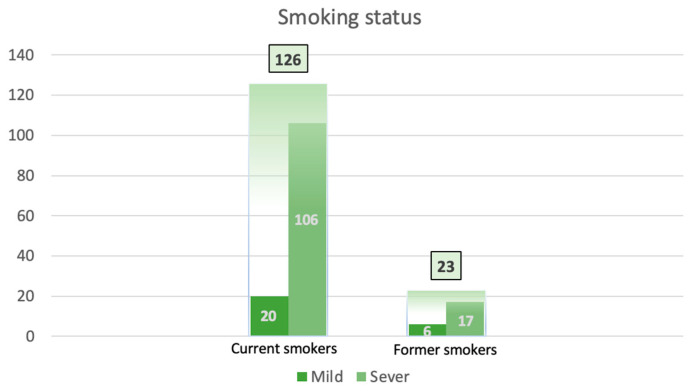
Smoking status.

**Figure 4 ijerph-20-04737-f004:**
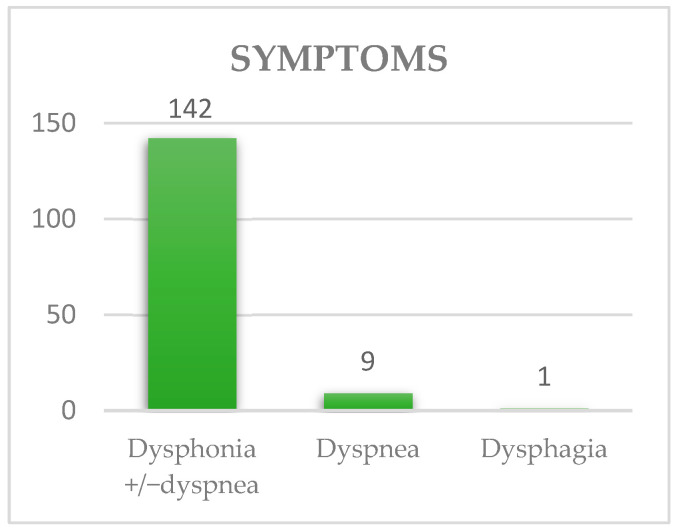
Symptoms associated with laryngeal cancer.

**Figure 5 ijerph-20-04737-f005:**
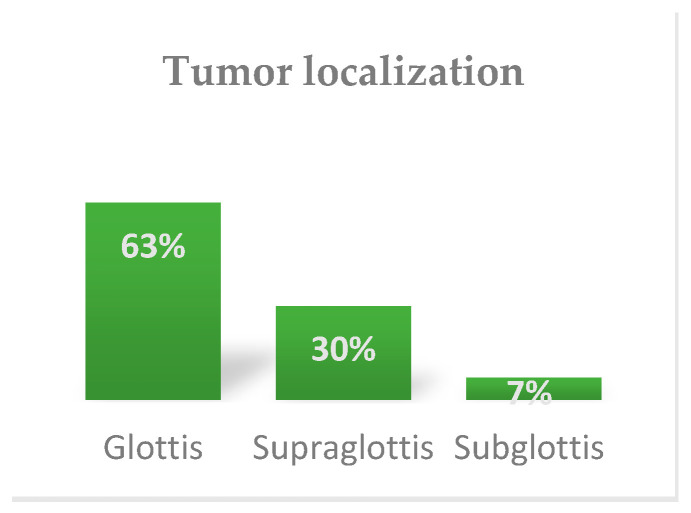
Tumor localization staging.

**Figure 6 ijerph-20-04737-f006:**
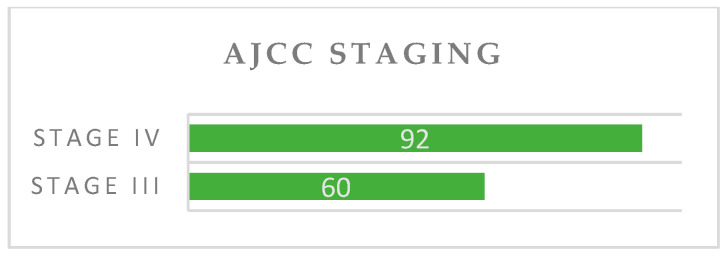
Tumor classification according to AJCC staging.

**Figure 7 ijerph-20-04737-f007:**
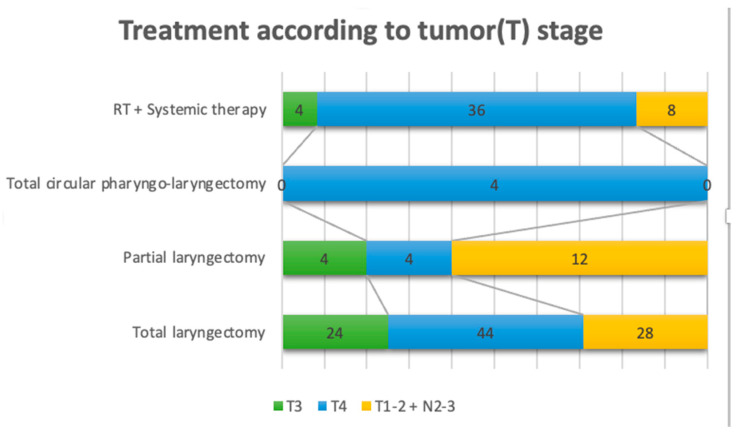
Treatment according to tumor stage.

**Figure 8 ijerph-20-04737-f008:**
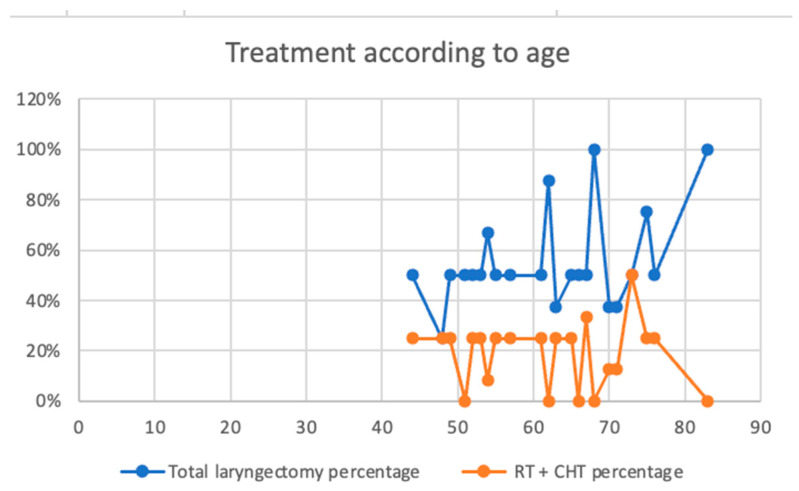
Treatment according to age.

**Figure 9 ijerph-20-04737-f009:**
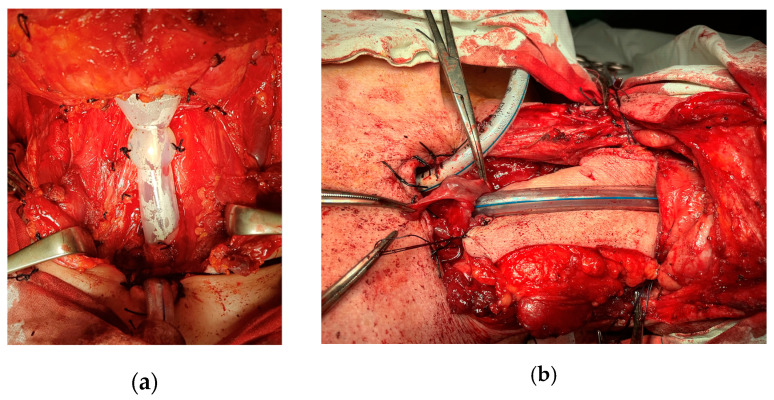
Intraoperative images. (**a**) Reconstruction of the pharynx with bypass salivary tube. (**b**) Reconstruction of the pharynx with myocutaneous major pectoralis muscle flap.

**Table 1 ijerph-20-04737-t001:** Romanian smoking rate [9].

Romanian Smoking Rate		
Year	Smoking Rate (Ages 15+)	Annual Change
2020	28.00%	−0.40%
2019	28.40%	0.00%
2018	28.40%	−1.30%
2015	29.70%	−1.70%
2010	31.40%	−1.80%
2000	35.00%	−1.80%

## Data Availability

Data are available on request due to privacy and ethical restrictions concerning patients’ personal data. The data presented in this study are available on request from the corresponding author since the main data are contained within the article.

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
