# Peer review of "Management of Locally Advanced Laryngeal Cancer—From Risk Factors to Treatment, the Experience of a Tertiary Hospital from Eastern Europe"

_ijerph, 2023, doi:10.3390/ijerph20064737_

Round 1
Reviewer 1 Report
The literature is full of recommendations on the approach to the treatment of laryngeal cancer. The American Society of Clinical Oncology gives us practical clinical instructions on the application of the Clinical Guideline for the Use of Larynx-Preservation Strategies in the Treatment of Laryngeal Cancer. According to those recommendations, N status significantly affects the selection of a surgical or organ preservation protocol. Did N status affect the choice of oncological treatment for patients with advanced cancer?T3 and T4 cancers are advanced, but the location of the cancer and the compartments affected by the tumor influence the choice of surgical option, reconstructive or total laryngectomy. To what extent did the localization and direction of tumor spread influence the choice of surgical technique? T4a carcinomas in the anterior paraglottic space do not exclude reconstructive surgery. What were the criteria for total laryngectomy? Did the patient's general status, comorbidities and pulmonary status influence the choice of surgical treatment?
How do you explain the high percentage of total laryngectomies if endoscopic surgery also includes some T3 cancers, and reconstructive surgery in addition to T3 refers to T4 cancers as well. The structure of treated advanced cancers according to TNM classification is not shown either.
Author Response
First of all, thank you very much for taking your time to review my article.
Here are my responses. Please let me know if I have to add something else.
-> To what extent did the localization and direction of tumor spread influence the choice of surgical technique? T4a carcinomas in the anterior paraglottic space do not exclude reconstructive surgery. What were the criteria for total laryngectomy?
How do you explain the high percentage of total laryngectomies if endoscopic surgery also includes some T3 cancers, and reconstructive surgery in addition to T3 refers to T4 cancers as well.
Response 1:
The reality of the Eastern European countries in oncological treatments is heavily influenced by:
- late diagnosis;
- high costs for oncological treatments, chemotherapy, radiotherapy;
- small number of new techniques for radiotherapy;
- small number of centers/hospitals and specialists;
- no protocols according to international standards approved by authorities.
This is why most of the surgeries are sometimes old fashion total laryngectomies with phonatory valve. Even with this vocal prosthesis which our clinic first introduced in the country (20 years ago), we have big resistance from patients and doctors (they do not want to use it, both patients and doctors).
I think these facts are important and probably I should introduce them in the discussion part. Please tell me your opinion. I really appreciate your help.
Tumor extension to the pharynx or esophagus required a total circular pharyngo-laryngectomy in 4 cases.
The criteria for total laryngectomy were: vocal fold fixation, invasion of the postcricoid area, invasion through the thyroid cartilage or invasion of the tissue beyond the larynx.
The majority of our patients are from groups of social risks, from villages and they do not go to general practitioner. The surveillance after larynx preservation treatment is difficult because a significant proportion of these patients do not present at the follow up.
-> Did N status affect the choice of oncological treatment for patients with advanced cancer?
Response 2:
All the patients from the study with clinically involved cervical nodes who were treated with surgery for the primary lesion underwent neck dissection. In patients who underwent primary non-surgical treatment neck dissection was not done. For the patients with pN2 or pN3 nodal disease concurrent radiotherapy and systemic therapy were performed after the surgical treatment.
-> Did the patient's general status, comorbidities and pulmonary status influence the choice of surgical treatment?
Response 3:
Yes, two patients from our study were not suitable for general anesthesia because of the comorbidities and they were referred to radiotherapy.
-> The structure of treated advanced cancers according to TNM classification is not shown either.
Response 4:
In the cases of partial laryngectomy 4 patients were T3, 4 patients T4 and 12 patients T1-T2 with N2-N3 nodal disease. For the patients that underwent total laryngectomy 24 patients were T3, 44 patients were T4 and 28 patients were T1-T2 with N2-N3 nodal metastases. The four cases with total circular pharyngo-laryngectomy were T4.
Reviewer 2 Report
Detection of early forms of laryngeal cancer is crucial for limited surgical intervention. Taking this into account, the authors conducted a two-year epidemiological study including 152 patients. These patients were eligible for surgical treatment at a single center.
In qualifying for treatment, the authors used their own modified UICC classification. The paper demonstrates the high surgical proficiency of the authors' team.
Comments:
1. in Fig.1, it seems unnecessary to emphasize the group second in size
2. please explain why a person who smoked only 100 cigarettes in his whole life was considered a cigarette smoker. While the WHO qualifies a smoker as a person who smokes 20 cigarettes a day, or millions of cigarettes in his lifetime?
3. were the patients included in the study after radiation therapy also title smokers or drank alcohol after this treatment? Or were they also tobacco and alcohol abstainers after RTX?
4. please provide the full name in place of the first use of the abbreviation e.g. GERD in the Results section.
5. was p16 or any other marker of HPV infection tested?
6. was the association of subsequent risk factors with clinical disease progression or TNM features tested? If not tested, kindly explain why this association should not be tested in this material.
7. Are patients asked about the use of electronic cigarettes?
Author Response
First of all, thank you very much for taking your time to review my article.
Here are my responses. Please let me know if I have to add something else.
- in Fig.1, it seems unnecessary to emphasize the group second in size
Thank you for your comment. Should I remove the Figure 1?
- please explain why a person who smoked only 100 cigarettes in his whole life was considered a cigarette smoker. While the WHO qualifies a smoker as a person who smokes 20 cigarettes a day, or millions of cigarettes in his lifetime?
The main classification of smokers that I used in our study (current, former and never smoker) is according to the definitions from the Centers for Disease Control and Prevention website, but I will also reminded the WHO classification, thank you for your comment. . It is my mistake that I did not insert the corresponding citation for the definitions.
- were the patients included in the study after radiation therapy also title smokers or drank alcohol after this treatment? Or were they also tobacco and alcohol abstainers after RTX?
I wasn’t thinking about controlling the tobacco and alcohol abuse after RTX, because all of the patients are instructed to quit both of them. Having in mind that they are from groups of social risk, from villages and they do not go regularly to general practitioners `I suspect that some of them are continuing. `Total laryngectomies patients in general makes smoking more difficult however.
If we consider their answers at regular visits they said quit smoking and drinking alcohol after starting radiation therapy. Questioning families, 9% of the patients continued to smoke and 6% of the patients continued to drink and 4% continued using both tobacco and alcohol.
- please provide the full name in place of the first use of the abbreviation e.g. GERD in the Results section.
Thank you for your suggestion. I modified.
- was p16 or any other marker of HPV infection tested?
No, it was not routinely tested.
- was the association of subsequent risk factors with clinical disease progression or TNM features tested? If not tested, kindly explain why this association should not be tested in this material.
I do not understand this question.
- Are patients asked about the use of electronic cigarettes?
The use of electronic cigarettes is not so common in Romania, and much more expensive, none of our patients are using this.
Reviewer 3 Report
Dear Authors,
Here are my detailed comments:
- Please describe more precisely the study population. For example I read "This retrospective study followed the patients diagnosed with larynx cancer" and then "The patients were classified in stage III and stage IV laryngeal cancer". Does this mean that only patients with advanced laryngeal cancer were included or that no patient with early laryngeal cancer came to your attention in the specified time span??
- Were also recurrences enrolled, or only primary tumors?
- Please insert citation for the sentence at lines 112-116
- Regarding alcohol consumption, please insert the number of drinks per day
- The TNM classification (both clinical and pathological) is completely missing, only the stage is included
- Did all 8 patients who underwent partial laryngectomy recur? And how is it possible that if the last patient was enrolled in 2022 the average time for recurrence was 2.5 years?
- Please pay attention to Figures' numbering (there are 2 Figure 4 and 2 Figure 5)
- Please do not repeat/describe the study results in the discussion section. Rather, use the Discussion to critically analyze and comment the results obtained.
- Conclusions are not supported by the results of Your study. In fact, this is only a descriptive study of a retrospective case review of a single center, and no statistical analyses have been performed. Therefore, it is not enough to establish causal relationships between risk factors and cancer development, or to identify epidemiological trends.
- Please pay attention to a few spelling errors. Furthermore, I suggest English language editing by a mother tongue speaker with expertise in scientific and medical writing.
Author Response
First of all, thank you very much for taking your time to review my article.
Here are my responses. Please let me know if I have to add something else.
- Please describe more precisely the study population. For example I read "This retrospective study followed the patients diagnosed with larynx cancer" and then "The patients were classified in stage III and stage IV laryngeal cancer". Does this mean that only patients with advanced laryngeal cancer were included or that no patient with early laryngeal cancer came to your attention in the specified time span??
My mistake I didn’t explain it more clearly.
Only patients with stage 3 or 4 laryngeal cancer that came to our hospital from January 2021 to December 2022 were included in our study.The advanced stage classification incorporated the cases with locally advanced tumor (T3 or T4) and the cases with early T classification (T1 or T2). We considered advanced the cases with T1 or T2 but with nodal disease N2-N3. Please advise if you think I have to use other criteria.
- Were also recurrences enrolled, or only primary tumors?
Both primary tumors and recurrences were enrolled in the study, this is why I decided to enroll 4 patients who came with recurrence after previous treatment. The treatment consisted of radiotherapy and chemotherapy in two cases and hemilaryngectomy followed by radiotherapy in the other two cases.
- Regarding alcohol consumption, please insert the number of drinks per day
Regarding alcohol use, the patients that recognised alcohol consumption were divided in three groups: light-to-moderate consumption, binge drinkers and heavy drinkers. From the total of 78.2% alcohol consuming patients, 11.5% consumed no more than one or two drinks per day, being included in the category of light-to-moderate alcohol consumers, 28% consumed 4 or more drinks on an occasion for women or 5 or more drinks on an occasion for men and were included in the binge drinkers group, whereas 38.7% consumed 8 or more drinks per week for women or 15 or more drinks per week for men and were included in the heavy drinkers group. The classification was made according to the definitions from the Centers for Disease Control and Prevention.
- The TNM classification (both clinical and pathological) is completely missing, only the stage is included
Thank you for your comment, please find below the detailed information:
- from the patients who underwent surgery:
Clinical: In the cases of partial laryngectomy 4 patients were T3, 4 patients T4 and 12 patients T1-T2 with N2-N3 nodal disease. For the patients that underwent total laryngectomy 24 patients were T3, 44 patients were T4 and 28 patients were T1-T2 with N2-N3 nodal metastases. The four cases with total circular pharyngo-laryngectomy were T4.
Pathological: The pathological TNM classification was: 38 patients pT4, 42 patients pT3, 34 patients pT2, 6 patients pT1. Regarding the nodal metastases 8 patients were classified as pN1, 43 as pN2 and 69 as pN3.
A discordance between cTNM and pTNM was observed in the cases with prior radiotherapy, the clinical T being overestimated. This could be because of fibrosis caused by radiation therapy and misinterpretation of computer tomograph images.
- from the patients who were RT
Clinical: The 4 patients referred to radiation therapy and chemotherapy were T3, 36 were T4 and 8 patients were T1 or T2 but with nodal metastases N2-N3.
- Did all 8 patients who underwent partial laryngectomy recur? And how is it possible that if the last patient was enrolled in 2022 the average time for recurrence was 2.5 years?
We enroll 8 patients who came with recurrence after previous treatment. Two of these patients had a previous partial laryngectomy and then underwent radiotherapy and systemic chemotherapy. Six patients only had concomitant radiotherapy and chemotherapy.
Yes, all these 8 patients with initial organ preservation treatment recurred, it is my mistake I didn’t explain in detail the fact that they were previously treated (before I started my retrospective study). So, I am talking about the patients that presented directly with recurrence, therefore patients who underwent surgery before 2021. This is way the average recurrence time was 2.5 years.
- Please do not repeat/describe the study results in the discussion section. Rather, use the Discussion to critically analyze and comment the results obtained.
Thank you for your comment. It is very useful for me.
In the discussion section I would like to add the following comments:
The reality of the Eastern European countries in oncological treatments is heavily influenced by:
- late diagnosis;
- high costs for oncological treatments, chemotherapy, radiotherapy;
- small number of new techniques for radiotherapy;
- small number of centers/hospitals and specialists;
- no protocols according to international standards approved by authorities.
This is why most of the surgeries are sometimes old fashion total laryngectomies with phonatory valve. Even with this vocal prosthesis which our clinic first introduced in the country (20 years ago), we have big resistance from patients and doctors (they do not want to use it, both patients and doctors).
The first symptoms are well-known by otorhinolaryngologists but ignored both by patient and primary care. If all specialties recognized first symptoms and knew the impact of the surgical and radiotherapy treatment, the disease could be better managed. There is a gap between internal medicine, other surgical specialties and general practitioners, who do not know the dimension of the oncological treatment for head and neck cancer as well as the impact on QOL. Utilizing accurate information from specialized clinics would have a major impact on diagnosing the disease in early stages, as well as developing screening programs, as there would be increased awareness of head and neck cancer. Taking into account the direct relationship between a risk factor and the impact of the disease, together with the existent information in the field from other European countries and globally, is key in developing good healthcare programs and prevention. This will decrease the costs of disease management as well.
A discordance between cTNM and pTNM was observed in the cases with prior radiotherapy, the clinical T being overestimated. This could be the result of fibrosis caused by radiation therapy and misinterpretation of computer tomograph images
The bypass tube reconstruction procedure was preferred due to the short surgical time, the shorter hospitalization stay and lower associated costs. It also allowed successful radiotherapy.
- Conclusions are not supported by the results of Your study. In fact, this is only a descriptive study of a retrospective case review of a single center, and no statistical analyses have been performed. Therefore, it is not enough to establish causal relationships between risk factors and cancer development, or to identify epidemiological trends.
Our center is the referral for oncological head and neck cancer in our country.
Unfortunately, we do not have a hospital statistician, so we decided to do a descriptive study.
I will reconsider the conclusions and I think important adding the next ones:
The impact of head and neck cancer is one of the largest in all malignancies. Knowing the peculiarities in different regions of the world will help in developing better prevention methods, improved screening programs, increase knowledge among healthcare professionals and in time, it will decrease the disease incidence as well as mortality. Early stage diagnosis is the only method for organ preservation and QOL.
Concluding the study, the reconstruction procedure with bypass salivary tube was applied to all patients due to the low costs and lower rate of post-radiotherapy complications that relate to nutrition. The main disadvantage is the impossibility of using the phonatory prosthesis for voice rehabilitation.
We found that although speech prostheses were inserted to almost all patients with total laryngectomy, 34% of them did not use it. The insurance system pays only for two prostheses per year, and sometimes a dysfunctional valve can lead to therapy failure. Social reintegration was much faster for patients who learned to use the phonatory prosthesis. The resumption of swallowing and phonation in partial laryngectomy cases was successfully completed in a maximum timeframe of six months, significantly longer compared to those who had total laryngectomy (11 weeks).
Round 2
Reviewer 2 Report
I see significantly improved manuscript.
Author Response
Thank you very much!
We appreciate the time and effort that you have invested into providing your valuable feedback on our manuscript.Reviewer 3 Report
Dear Authors,
Thank You for Your nice work.
Unfortunately, I think Your manuscript is still too broad and descriptive. The article is not well addressed to investigate on Your study question, which must be more specific and clear from the beginning. Conclusions are not supported by the results of Your specific work.
I suggest to focus on a more specific subject and use Your data to address Your study objectives.
